# Polyvinylalcohol-carbazate (PVAC) reduces red blood cell hemolysis

**Felix Sellberg**[1], **Fanny Fredriksson**[2], **Thomas Engstrand**[3], **Tim Melander Bowden**[4], **Bo Nilsson**[1], **Jaan Hong**[1], **Folke Knutson**[1], **David Berglund**[1]*

**1** Department of Immunology, Genetics and Pathology, Uppsala University, Uppsala, Sweden, **2** Department of Women's and Children's Health, Section of Pediatric Surgery, Uppsala University, Uppsala, Sweden, **3** Department of Surgical Sciences, Uppsala University, Uppsala, Sweden, **4** Department of Chemistry—Ångström Laboratory, Uppsala University, Uppsala, Sweden

* david.berglund@igp.uu.se

**Data Availability Statement:** All relevant data are within the paper and its Supporting Information files, including gel images in their unaltered form.

**Funding:** The authors received no specific funding for this work.

## Abstract

### Background and objectives

The objective of this study was to investigate whether a soluble polymer and aldehyde-scavenger, polyvinylalcohol-carbazate (PVAC), can inhibit hemolysis in the storage of red blood cells (RBC).

### Study design and methods

The effect of PVAC was assessed over a wide range of concentrations, using absorption spectroscopy to evaluate the level of hemolysis. Moreover, osmotic stability and aldehyde-scavenging potential of RBC were assessed after storage in PVAC.

### Results

After test tube storage for two weeks, red blood cell hemolysis was lower with PVAC compared to controls (mean difference 23%, 95% CI 16–29%, p < 0.001). A higher level of hemolysis led to a pronounced effect with PVAC. RBC stored in PVAC improved both the binding of free aldehydes (p <0.001) and the osmotic stability (p = 0.0036).

### Conclusion

Erythrocytes stored with PVAC showed less hemolysis, which might be explained by the ability of PVACs to stabilize the cell membrane and decrease oxidative injury.

## Introduction

Hemolysis is in most areas of medical science unwanted and it can be divided into hemolysis *in vivo* or *in vitro*.

Hemolysis *in vivo* occurs in a variety of diseases and can have both intrinsic and extrinsic causes. Intrinsic causes are either structural or functional defects in the red blood cells (RBCs)

**Competing interests:** DB, FK, TE and TB have applied for patent protection using PVAC to decrease hemolysis during erythrocyte storage. The patent is filed but is not yet public and thus is not assigned a searchable identifier. TE, DB and TB are shareholders in a company, PVAC Medical Technologies LTD (PMT), developing PVAC for clinical applications. TE is an employee of PMT. DB and TB have acted as consultants towards PMT. The other authors have no conflict of interest. This does not alter our adherence to PLOS ONE policies on sharing data and materials.

which explains diseases such as thalassemia, sickle-cell disease or enzyme deficiencies. Extrinsic causes occur in the external RBC environment, e.g. autoimmune hemolytic anemia, where antibodies directed towards RBCs are formed or infections causing hemolytic uremic syndrome where toxins lyse RBCs [1]. The treatment of these diseases would be to limit the hemolysis in patients, combined with blood transfusions. However, antihemolytic agents are not available on the market [2].

In vitro, up to 8% of all blood samples taken at an emergency department are affected by hemolysis. Hemolysis is the main (60%) cause of failed laboratory tests of blood samples from the clinic, which leads to the need for repeated blood samples from patients, more staff time, and increased economic costs [3,4]. Hemolysis also interferes with test results, most commonly e.g. changes in potassium levels and immunological assays [5].

The storage of RBCs for use in clinical blood transfusions may be decreased by hemolysis [6,7]. The US FDA and European guidelines for transfusion state that hemolysis should be <1% and 0.8%, respectively, and that more than 75% of transfused RBCs remain in circulation 24 h after transfusion. Using contemporary storage solutions RBCs are usually able to meet these criteria for storage times for up to 42 days.

Protein carbonylation refers to a type of protein oxidation usually started by an increase in production of reactive oxygen species, which in turn starts an oxidation cascade counteracted by reducing systems such as glutathione [8]. However, if these systems are overwhelmed the reaction causes irreversible downstream modifications of proteins disrupting their function [9]. RBCs are highly resistant to oxidative stress but the systems are linked to metabolic status (glucose availability) of the RBC [10,11]. Oxidative injury contributes to the aging and destruction of RBC, which is further compounded by glucose depletion [7,12].

There is a need for novel exogenous reductive agents. Polyvinylalcohol-carbazate (PVAC), a polymeric compound that is highly soluble in aqueous solutions, has the capacity to bind endogenous aldehydes and neutralize oxidative stress. The objective of this study was to investigate PVAC's ability to inhibit hemolysis in the storage of RBCs.

## Materials and methods

### Preparation and characteristics PVAC

PVAC was manufactured at Department of Chemistry—Ångström Laboratory, PVAC was manufactured at Department of Chemistry—Ångström Laboratory, Uppsala University, Sweden. PVAC is a 15- to 35-kDa polymer composed of a polyvinylalcohol (PVA) backbone that has been postmodified to partially include carbazate groups. The hydrazine moiety of the carbazate group is nucleophilic and reacts with electrophiles such as carbonyls (aldehydes or ketones) to form Schiff base like carbazones (Fig 1). PVAC has reactivity towards several electrophilic compounds such as aldehydes, carbonyls and ROS. Freeze-dried PVAC was dissolved in physiological saline (0.9% NaCl), vortexed for 30 s and used within 1 h of reconstitution.

### Preparation of RBC

For experiments investigating RBC fresh erythrocyte concentrates in Sagman solution were used. PVAC was dissolved in 0.9% NaCl and added to the erythrocyte concentrates. Whole blood was used for experiments examining biocompatibility. Both fresh erythrocyte concentrates and whole blood were obtained from healthy blood donors at the Uppsala University Hospital Blood Bank. A healthy blood donor is a member of the public that has passed the process to become a regular blood donor at the Blood Bank. This is a rigorous process that includes medical and laboratory assessments that determine the health status of the donor and that follow internationally established standards. [13]

**Fig 1. The chemical structure of PVAC.** Polyvinylalcohol-carbazate (PVAC) condensation reaction with aldehyde at neutral conditions leads to the formation of a stable carbazone adduct and a water molecule. Unmodified repeat units of PVA are denoted with n and carbazate groups conjugated to repeat units are denoted by m. The level of substitution of PVA with carbazate groups is about 10% (n = 0.9; m = 0.1).

## Storage of RBC in test tubes

Erythrocyte concentrates were split into 5-mL portions. PVAC was dissolved in NaCl (250 μL) and added to the erythrocytes over a broad range of concentrations. As control, NaCl alone was added at a volume equivalent to that of the PVAC solution. Erythrocyte concentrates were stored in cold (4˚C) or room temperature (20˚C) for 2 weeks in polypropylene test tubes. For erythrocytes stored at room temperature more PVAC was added after one week because the hemolysis was expected to be greater in warm compared to cold storage. During contemporary storage of RBC the temperature is lowered to stop the metabolism and increase pH during storage. [14] Experiments in cold storage were done with six donors in duplicates and experiments in warm storage were done with samples from two donors in duplicates. Supernatant samples were regularly collected from the erythrocyte concentrates after centrifugation (2000 x g for 10 m). Free Hb was measured with the plasma-low Hb method (Hemocue, Angelholm, Sweden). The percentage of hemolysis was calculated by dividing the supernatant concentration by the total hemoglobin concentration for each sample.

## Storage of RBC in 96-well plates

Erythrocyte concentrates were diluted in of NaCl (v/v 100/100 μL) with or without additional PVAC, PVA (Sigma) and ethyl-carbazate (EC) (Sigma). The final volume of 200 μl was added to a round bottom 96-well cell culture plate with a lid, allowing oxygen exchange, and stored for one month at 4˚C. Plates were assayed for hemolysis at day 0, 4, 7, 14, 21 and 28. At analysis the plate was spun down (2000 g x 10 min) and supernatant (100 μL) was analyzed for absorbance (540 nm) using a Synergy HTX plate reader (BioTek). The 540-nm wavelength was used because it represents the absorbance peak for the oxygenated state of hemoglobin, the most common state during storage.

## Assessment of hemoglobin levels

To exclude any direct interference of PVAC with the detection of hemolysis, whole blood was centrifuged (2000 x g for 10 min) and the erythrocyte portion was lysed using ethanol and recentrifuged. Free Hb was obtained and dissolved in PBS to a concentration of 10 g/L. PVAC was added to the free Hb and incubated for 2 h before centrifugation (2000 x g for 10 min). Free Hb was measured with the plasma-low Hb method (Hemocue). The spectrum was also

analyzed using a Synergy HTX reader (BioTek) measuring absorbance between 300 and 700 nm at 10-nm increments.

## Red blood cell fragility testing

RBC from two donors in duplicates were spun down and resuspended in NaCl solution. PVAC was dissolved in NaCl and then added to the red blood cells at different concentrations. The same volume of NaCl was added to the control. The mixture was left to incubate for 1 h. Deionized water (DI water) was then mixed with 0.9% NaCl (v/v 1:1) to create a hypotonic solution. Red blood cells were incubated in the hypotonic solution for 10 min. Erythrocytes were finally spun down and the supernatant sampled and analyzed for absorbance at 540 nm using a TECAN sunrise plate reader (Tecan Trading AG, Switzerland).

## Aldehyde resistance in test tubes

After cold storage, erythrocyte concentrates from four donors in duplicates were tested for aldehyde resistance. Erythrocytes (0.9 mL) were exposed to acetaldehyde (0.1 mL, 100 μg/mL, solved in 0.9% NaCl), at a final concentration of 10 μg/mL acetaldehyde in the erythrocyte suspension. After 60 min of incubation under stirring, the erythrocyte concentrate was centrifuged (2000 x g for 10 min). Supernatants were removed and used to determine aldehyde scavenging potential, i.e. resistance to the addition of acetaldehyde. Free aldehydes were measured using the Megazyme Acetaldehyde Assay Kit (Bray, Ireland), according to manufacturer's instructions for microplate technique using 100 μL as sample volume. The plates were read at 340 nm using Synergy HTX plate reader (BioTek, Winooski, VT, USA) before and 10 min after adding aldehyde dehydrogenase. Results are reported as the change in absorbance at 340 nm.

## Whole blood chamber model

Chambers, composed of a polymethylmethacrylate (PMMA) microscope slide with two PMMA rings affixed acting as wells, were filled with whole blood from three donors and hermetically covered. Chambers, pipettes, and test tubes were heparinized prior to use in the experiments according to manufacturer's protocol (Corline Biomedical, Uppsala, Sweden). Whole blood (1.5 mL) was pipetted into the heparinized chambers and PVAC dissolved in PBS (50 mg/mL, 75 μL) was added to bring the final concentration to 2.5 mg/mL within the chamber. In control chambers, PBS (75 μL) was added. A potently thrombogenic metal (titanium disc) was used as a positive control; the disc was placed as a lid to the chamber in lieu of heparinized plastic normally used, leading to an activation of both complement and coagulation. The chambers were placed on a mechanized wheel spinning at 20 rpm for 2 h, as previously described [15]. Blood was sampled for total platelet count (TPC) using a Sysmex XP-300 hematology analyzer (Kobe, Japan) directly after the experiment and centrifuged at 400g for 10 min. The plasma was frozen and stored at -70˚C. Complement and coagulation markers were assessed with ELISA.

To detect thrombin-antithrombin (**TAT**) complexes, an antihuman thrombin antibody (Enzyme Research Labs Inc., USA) was used as the capture antibody. Bound TAT was detected with horseradish peroxidase (HRP)-conjugated antihuman antithrombin antibody (Enzyme Research Labs Inc., USA). Pooled human serum diluted in EDTA plasma was used as reference standard.

C3a is a soluble protein formed via protein cleavage during complement activation and it acts as a proinflammatory signaling molecule. To detect **C3a**, an anti-C3a monoclonal antibody 4SD17.3 (produced in house) was used as capture antibody. Bound C3a was detected

with biotinylated anti-C3a antibody followed by HRP-conjugated streptavidin (GE Healthcare, Sweden). Zymosan-activated serum (ZAS), calibrated against a solution of purified C3a, was used as reference standard.

To detect sC5b-9, also known as terminal complement complex (**TCC**), the anti-neoC9 monoclonal antibody aE11 (Diatec Monoclonals AS, Oslo, Norway) was used as capture antibody. Bound sC5b-9 was detected by biotinylated polyclonal anti-C5 antibody (Biosite), followed by HRP-conjugated streptavidin (GE Healthcare). ZAS was used as reference standard. Absorbance was measured at 450 nm using a TECAN Sunrise microplate reader (Tecan Trading AG).

In the model, platelet consumption and increased levels of circulating TAT are interpreted as an increased activity of the coagulation system while increasing levels of C3a and TCC are interpreted as increased activity in the complement system.

### Chemical modification of PVAC

To visualize PVAC with flow cytometry and to perform immunoprecipitation assays a chemical modification was done with the carbazate groups and their reactivity towards electrophilic compounds. Fluorescein isothiocyanate (FITC) is a fluorochrome reactive towards most nucleophilic substances including amines and was therefore used to conjugate PVAC [16]. For immune precipitation, conjugation to biotin was used to create a means of extracting PVAC from a solution. Substituting carbazate groups on PVAC can theoretically affect the solubility as well as the scavenging potential of PVAC. To show the conjugation of PVAC to FITC or biotin did not have these effects, the levels of substitution were determined (supporting information, S1 File and S1 and S2 Figs). A low level of substituted carbazate groups was used (1%) which did not affect either solubility or scavenging characteristics.

In brief, PVAC was dissolved in deionized water (100 mg, 5 mL). FITC (Sigma, Upplands Vasby, Sweden) was dissolved in DMSO (WAK-Chemie, Steinbach, Germany) (1 mg, 0.1 mL). The components were mixed and allowed to react during 2 h under stirring (magnetic set at 200 rpm) in a glass beaker covered with foil. After 2 h the components were transferred to dialysis tubing, 3500 MWCO (Thermofisher), allowing unbound FITC to dissipate leaving only FITC conjugated to PVAC in the tubing. The tubing was placed in a large glass beaker filled with deionized water (DI water: reaction fluid, 1000:1). The beaker was covered with foil and set to rest overnight at room temperature under stirring (magnetic set at 200 rpm). The fluid inside the dialysis tubing was then transferred to a polystyrene tube (15 mL) and freeze dried. After freeze drying the compound was stored at -20˚C until use.

To conjugate to biotin the ImmunoProbe™ Biotinylation Kit (Sigma) was used, according to manufacturer's instructions, with the volumes adjusted for the desired conjugation level (1%). After conjugation the Biotin-PVAC was purified with dialysis as outlined above. The product was then freeze dried and stored at -20˚C until use.

### Flow cytometry

RBCs were incubated with FITC-PVAC (100 μL, 1 mg/mL dissolved in PBS) and then washed twice in PBS before analysis. In addition, RBC were incubated in either isotonic or hypotonic NaCl (0.9–0.5%) solutions with added FITC-PVAC (1 mg/mL) for 30 min with shaking in the dark. After incubation RBCs were washed twice before analysis using an Accuri C6 flow cytometer (BD, Franklin lakes, NJ, USA). Data processing was done in FlowJo (FlowJo LCC, Ashland, OR, USA). RBCs were identified using FSC and SSC and mean fluorescent intensity (MFI) was recorded for FITC.

### Immunoprecipitation (IP) and mass spectrometry (MS)

For detailed description of IP and MS please see the supporting information. (S1 File)

### Statistical analysis

All data handling and statistics were done in Graphpad prism (GraphPad Software, Inc, La Jolla, CA, USA). Significant p values are defined as <0.05 and denoted by *. All graphs are presented with mean values and error bars as SEM. Initially Shapiro-Wilk normality test was done to determine the nature of the data. In experiments with two groups, t-tests or nonparametric counterparts were used, whereas experiments with more than two groups were analyzed by ANOVA. When experiments had two parameters, e.g. hemolysis and time, two-way ANOVA was done. In experiments with only one time point, or time-independent observations such as hemolysis after two weeks of storage, one-way ANOVA was done.

### Ethical permits

The study was conducted within the quality system of the Uppsala University Hospital Blood Bank after a decision in the Department Board. Experiments using whole blood were approved by the local Swedish ethical committee in Uppsala, with support of the Swedish law (6§ 2003:460) regulating ethical approvals of research. The specific decision DNR 2008/264 has been appended to the supplementary information.

## Results

### Storage in test tubes

After storage of RBC from six donors in test tubes for two weeks at 4°C, a dose-dependent decrease in hemolysis was observed with additional PVAC. (Fig 2A) Hemolysis was lower with the addition of 2.5 (0.25%, 0.11–0.35, p = 0.0048) and 0.5 (0.17%, 0.01–0.33, p = 0.0478) mg/mL of PVAC compared to the control. This preventive effect on hemolysis led to further investigations using RBC duplicates from four donors, these were used an put under additional stress with an increase of temperature during storage.

Blood storage in test tubes showed as expected an increase in hemolysis at room temperature (3.1% vs 1.08%, p = 0.0041) compared to cold storage, at 4°C. During the first week of storage, the groups did not differ significantly but as hemolysis increased during the second week there was a protective effect using PVAC. (Fig 2B) Hemolysis was lower in all groups with added PVAC, 2.5 (1.75%, p < 0.001), 0.5 (1.86%, p < 0.001), 0.1 (1.87%, p < 0.001), 0.01 mg/mL (2.15%, p < 0.001). The most effective prevention was seen when PVAC 2.5 mg/mL was used, with a 44% reduction in hemolysis compared to the control.

Hemoglobin was measured before and after adding the second dose of PVAC to exclude any interaction between hemoglobin and PVAC.

Levels of free hemoglobin did not differ. Moreover, the absorbance spectrum of hemoglobin was analyzed between 300 and 700 nm, with and without additional PVAC, showing no difference in the spectra. Please see supporting information for data. (S5 Fig)

### Red blood cell osmotic fragility and flow cytometry

RBCs from four blood donors exposed to a hypotonic solution were rapidly lysed. However, when red blood cells were treated with PVAC prior to exposure, a dose dependent reduction of hemolysis was seen. Incubation with PVAC at 2.5 mg/mL reduced hemolysis with a mean of 63%, 56–71, p = 0.0026, when compared to the control. (Fig 3A) No apparent binding was observed when RBCs from three different blood donors were mixed with FITC-PVAC in

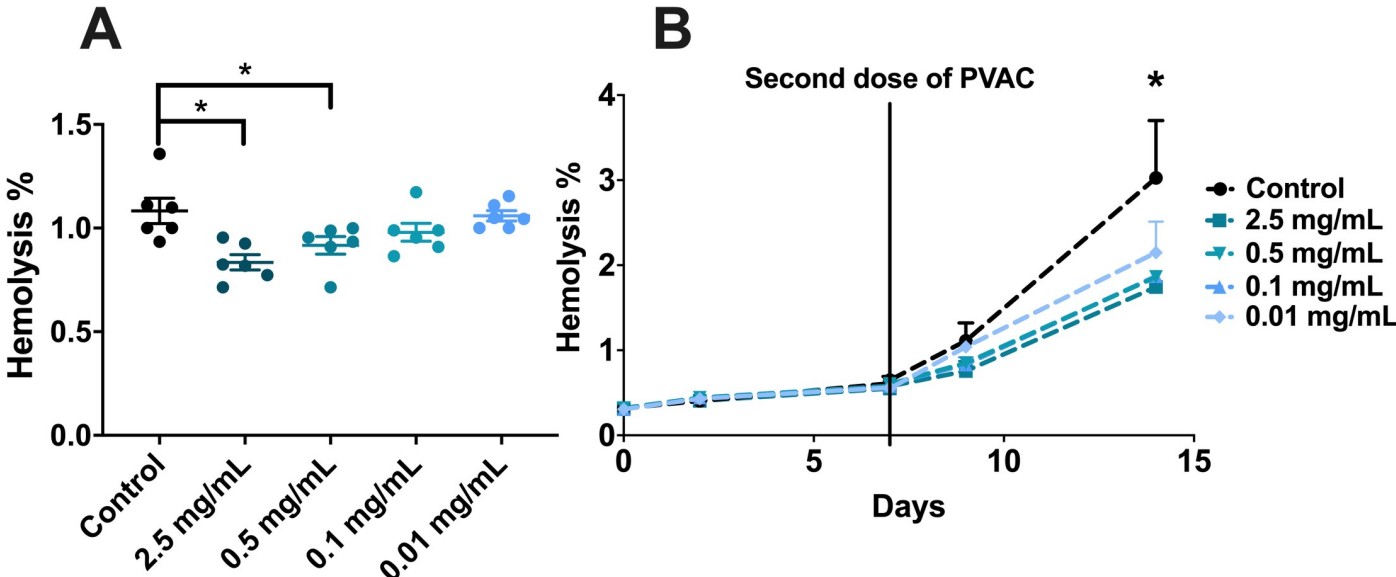

**Fig 2. RBC stored in test tubes with PVAC added to the solution.** (a) RBC stored in test tubes for two weeks at 4˚C, a dose-dependent reduction of hemolysis was seen with increasing doses of PVAC. 2.5 (0.25%, 0.11–0.35, p = 0.0048) and 0.5 (0.17%, 0.01–0.33, p = 0.0478) mg/mL of PVAC compared to the control. **(b)** RBC stored in test tubes for two weeks at room temperature (20˚C), where hemolysis was measured three times a week. A second dose of PVAC or NaCl was added after one week. Mean hemolysis in the control after two weeks was 3.1% and the addition of PVAC lowered hemolysis in all groups: 2.5 (1.75%, p < 0.001), 0.5 (1.86%, p < 0.001), 0.1 (1.87%, p < 0.001), 0.01 mg/mL (2.15%, p = 0.001).

isotonic NaCl or PBS. However, in a hypotonic solution an increase in binding was observed. MFI for cells in hypotonic solution was increased, 1021 vs 335 (mean difference 686, 389–983, p = <0.001). (Fig 3B) Altered concentrations of NaCl in the hypotonic solution did not differ.

### Aldehyde resistance after storage

Aldehyde levels 60 min after addition of acetaldehyde showed that RBC from four donors stored in PVAC were more efficient to neutralize acetaldehyde. The effect was dose dependent. Addition of 2.5 mg/mL PVAC reduced the amount of free aldehydes by 55% (p = <0.001). (Fig 4)

### Blood compatibility

Blood compatibility was studied in a whole blood model where whole blood from three circulated in chambers for 2 h with adding either PVAC, a stimulating material (thrombogenic metal) or NaCl and subsequently sampled for analyses. PVAC did not activate the coagulation system as levels of TPC and TAT were comparable to the control. The complement cascade, assessed by TCC and C3a, was similarly unaffected compared to the control. (Fig 5)

### Mass spectrometry

PVAC conjugated to biotin associated with beads showed no difference in the gel pattern or in the final MS analysis between the proteins isolated with or without additional PVAC. For detailed results please see the supporting information. (S1 File and S3 and S4 Figs)

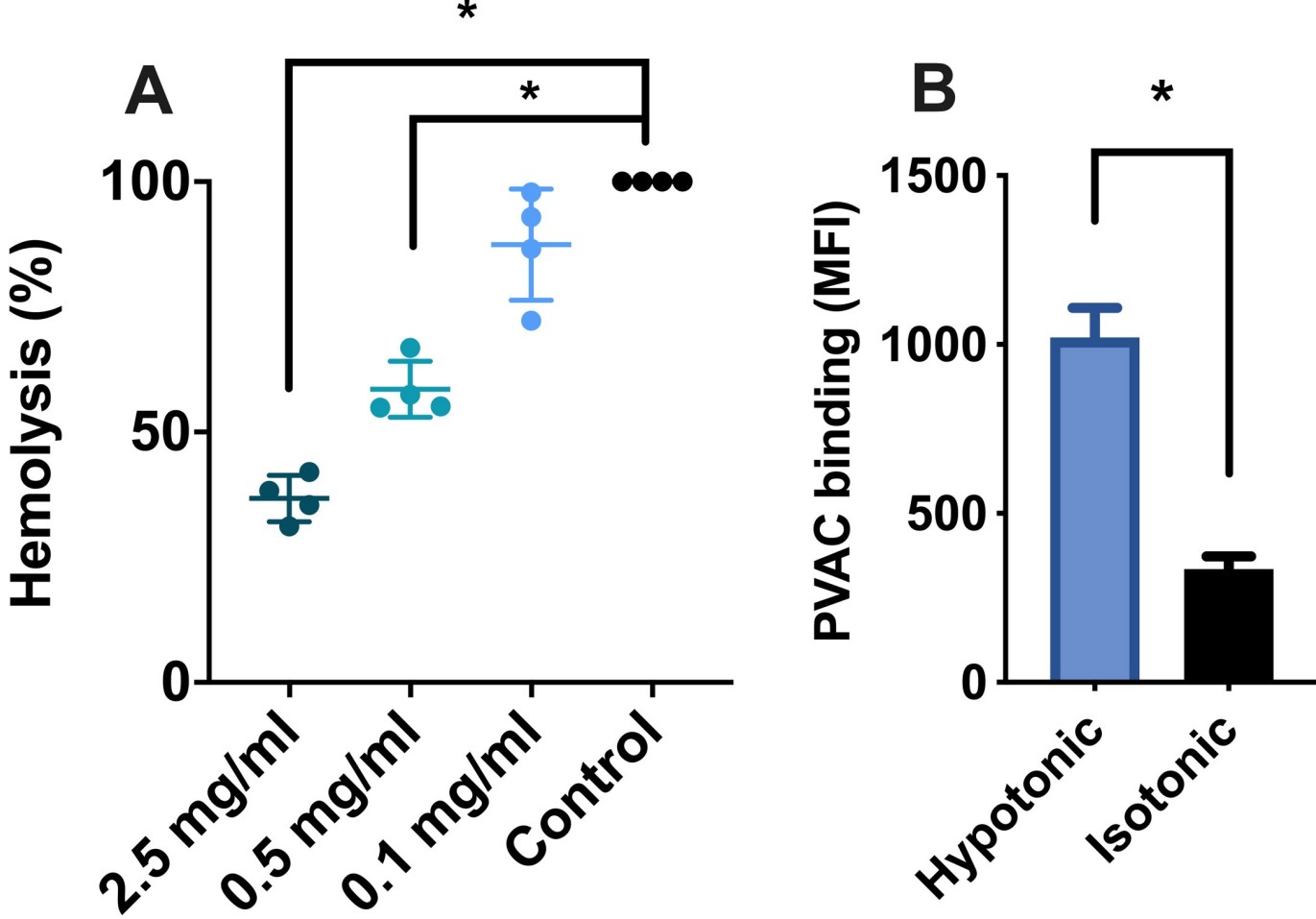

**Fig 3. Osmotic stability of RBC stored with PVAC and association studied with flow cytometry.** (a) RBC were exposed to a hypotonic solution and sampled for free hemoglobin. Hemolysis was measured as absorbance at 540 nm, with values normalized to the control. Addition of PVAC resulted in a dose-dependent reduction of hemolysis, with a mean reduction of 63% for PVAC 2.5 mg/mL compared to the control (p = 0.0026). (b) RBC incubated with FITC-conjugated PVAC in a hypotonic or isotonic solution. The MFI was higher when the RBC were incubated in a hypotonic solution compared to an isotonic solution, mean difference 686 (389–983, p = <0.001).

### Influence of chemical components of PVAC

RBC from four donors stored in the presence of a low-molecular-weight carbazate (ethyl-carbazate, EC), PVA backbone and, PVAC displayed different patterns of hemolysis over time. During the first few days of storage the groups did not differ, but after 14 days all but the lowest concentration of EC (2.5 mg/mL, 500 and, 100 μg/mL) had a higher level of hemolysis compared to the control. (Fig 6A) After one month storage of RBC (from eight different blood donors), all concentrations of EC (2.5 mg/mL, 500, 100 and, 20 μg/mL, 288.5, 257.8, 120.5 and, 24.9% mean increase, respectively) and the lowest concentration of PVA (20 μg/mL, 34.1% mean increase) had a higher level of hemolysis compared to the control. All concentrations but 20 μg/mL of PVAC lowered the level of hemolysis (2.5 mg/mL, 500 and, 100 μg/mL, 39.6, 22,1 and, 19.6% mean decrease, respectively). (Fig 6B)

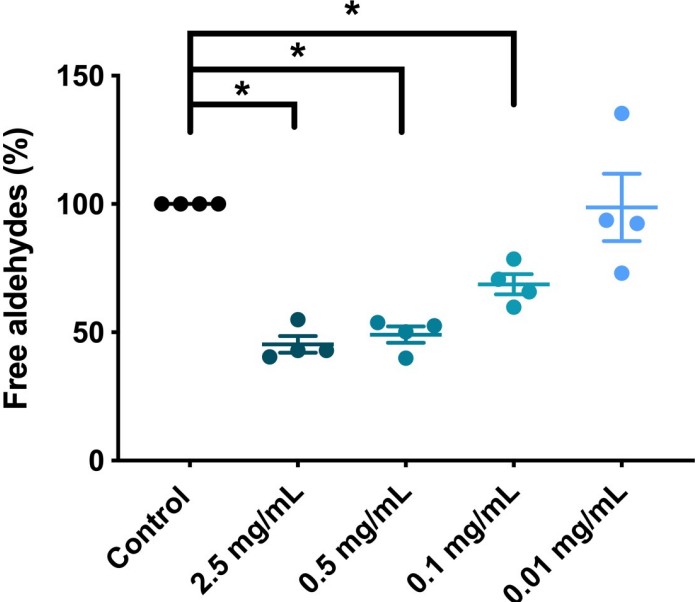

**Fig 4. Aldehyde resistance after storage with PVAC.** RBC stored in different concentrations of PVAC were exposed to acetaldehyde. The level of free aldehydes was measured after 60 min as absorbance at 340 nm, with values normalized to the control. Addition of PVAC prior to storage resulted in a dose-dependent decrease in the level of aldehydes compared to the control.

## Discussion

This is a proof-of-concept study evaluating the potential of PVAC to reduce hemolysis in different settings. Storage in test tubes demonstrated a dose-dependent effect with additional PVAC, resulting in lower levels of hemolysis. When RBCs are put under additional stress with increased storage temperature, PVAC decreased the hemolysis by about one half. PVAC also

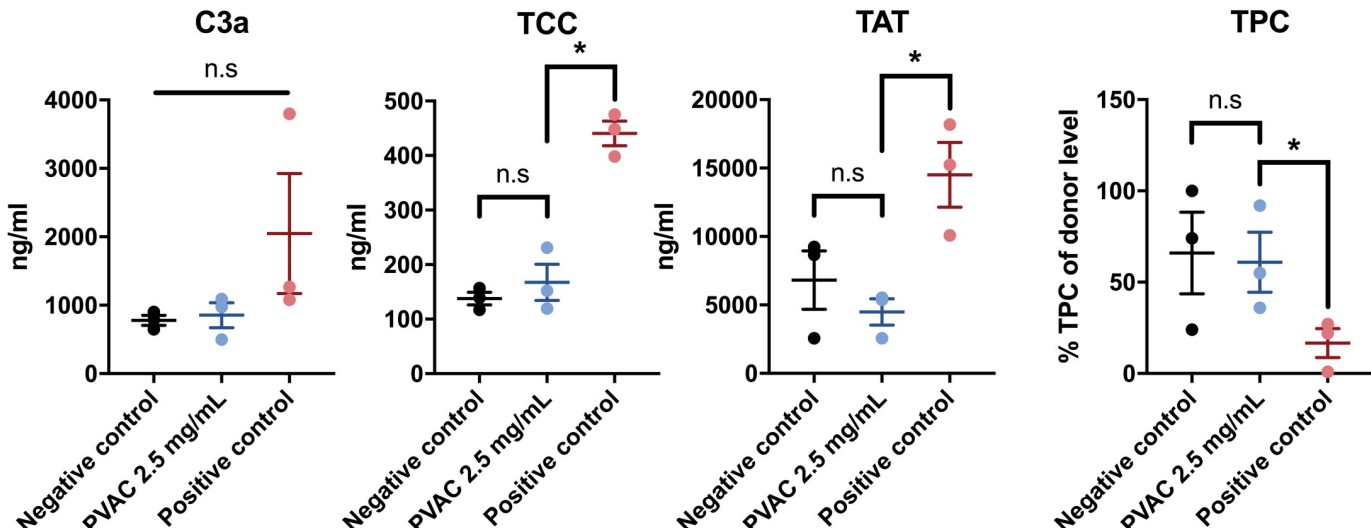

**Fig 5. Blood biocompatibility of PVAC in a whole blood model.** Graphs showing the levels of coagulation markers TPC and TAT and complement activation markers TCC and C3a. No difference between PVAC and the negative control was found, supporting biocompatibility without activation of complement or coagulation.

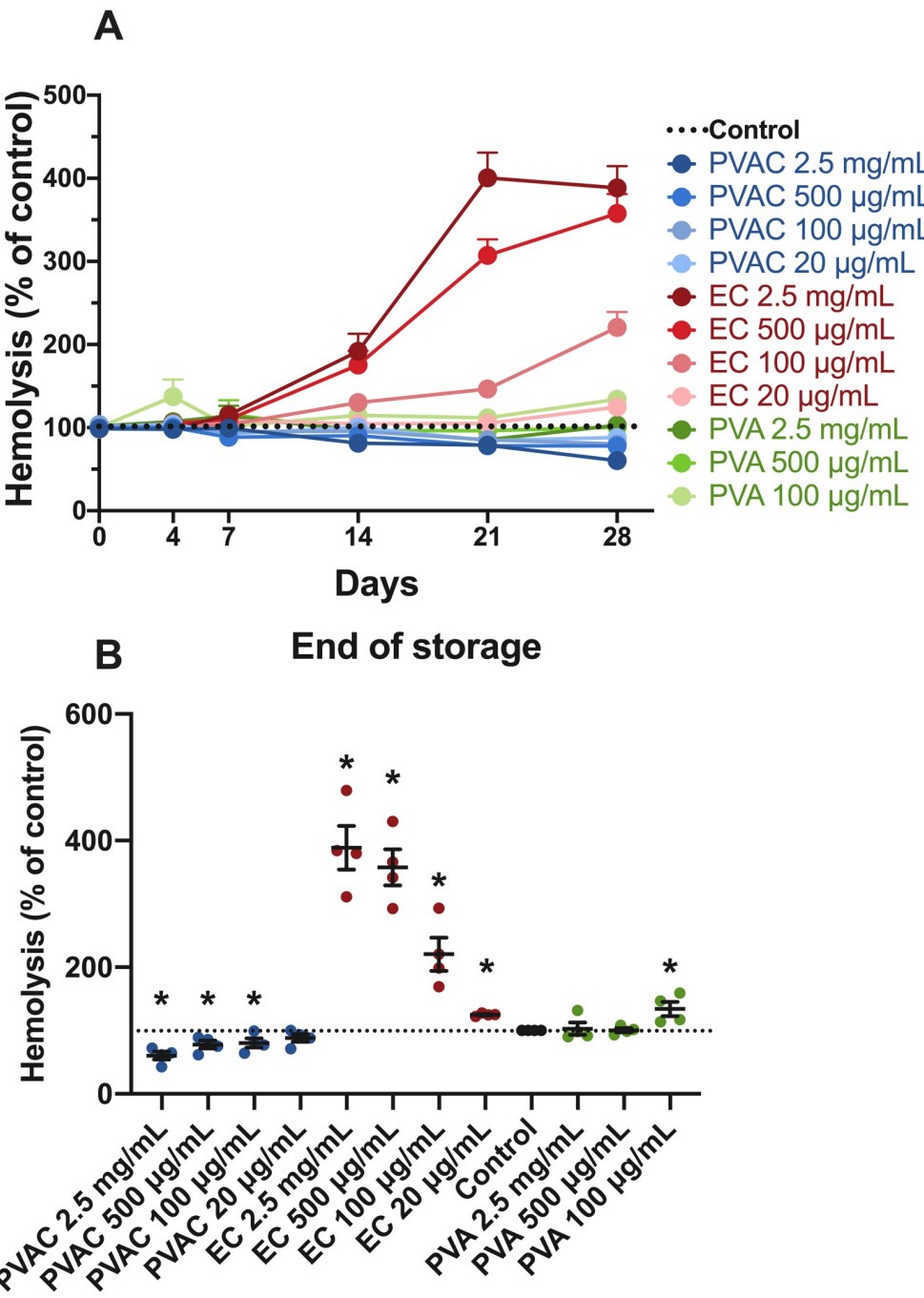

**Fig 6. Influence of chemical components of PVAC during storage of RBC.** Comparison between PVAC, EC and PVA during storage of RBC. In **(a)** all groups are plotted with change in levels of hemolysis over time, EC starts to deviate from the control after 7 days of storage and all EC groups had higher levels of hemolysis compared to the control. In **(b)** values after 1 month (end of storage) are shown: PVA at 100 µg/mL increased hemolysis while PVAC at 2.5 mg/mL, 500 and, 100 µg/mL decreased the levels of hemolysis compared to the control.

conferred proportionally higher protection of hemolysis at room temperature compared to cold storage, possibly signifying that there is an increased stress to be neutralized by PVAC at

room temperature and that PVAC can act as an inhibitor when additional stress is put on the cells.

One of the factors contributing to the storage lesion is oxidative injury, [17] with previous studies linking increasing concentrations of malondialdehyde (MDA) to the progress of the storage lesion. [12] RBC stored in PVAC exhibited an increased aldehyde binding capacity, which could help blood storage by preventing oxidative by-products from injuring the RBC. In addition, when red blood cells were tested for osmotic fragility after exposure to PVAC, the level of hemolysis decreased substantially. PVAC thus seems to also provide a membrane stabilizing effect, making erythrocytes more resistant to lysis with hypotonic solution. When analyzed with flow cytometry the signal for RBC increased more than threefold when the cells were in hypotonic solution. This along with the membrane stabilizing effect indicate that PVAC associates directly with RBC during stress. When separately assessing the different components of PVAC, a low-molecular-weight carbazate (ethyl-carbazate) negatively affected storage resulting in an increase of hemolysis after storage for one month. The backbone used in PVAC synthesis, PVA, also resulted in an increase in hemolysis after one month of storage at certain concentrations. Still, this effect was not as pronounced as that of ethyl-carbazate. Indeed, PVA seemed rather inert in comparison, which is not unexpected considering its widespread use in materials commonly used in medicine and elsewhere. It is therefore not the individual chemical properties of either the carbazate moiety or the PVA backbone alone that are responsible for the observed effects but rather their combination that results in properties that are able to reduce hemolysis.

Based on the present study, PVAC seems to be a promising agent worthy of further investigation for its ability to reduce hemolysis. So far, no *in vivo* toxicity has been observed with PVAC, even in very high concentrations administered intravenously in rodents (unpublished data). Future studies will be aimed at assessing the biodistribution and clearance after *in vivo* administration of PVAC to approach clinical applications. Whereas the best potential clinical use for PVAC remains to be determined, the broader concept of neutralizing aldehydes and other reactive components through activated polymers serves as a proof-of-concept for a novel group of biologically active substances.

## Supporting information

**S1 Fig. Solubility of PVAC with varying degrees of conjugation.**
(TIF)

**S2 Fig. Ligand binding capabilities with varying degrees of conjugation.**
(TIF)

**S3 Fig. Association of biotin-PVAC with magnetic anti-biotin beads.**
(TIF)

**S4 Fig. Gel picture of bands isolated via IP.**
(TIF)

**S5 Fig. Abosorbance spectra of hemoglobin in the presence of PVAC.**
(TIF)

**S6 Fig. Raw image file for gel picture.**
(TIF)

**S1 File. Detailed description of experiments, in addition results from MS and IP.**
(DOCX)

**S2 File. Raw data file.**
(XLSX)

**S3 File. Raw data from MS, PVAC.**
(XLSX)

**S4 File. Raw data from MS, control.**
(XLSX)

## Acknowledgments

We would like to thank Uppsala University Hospital Blood Bank for their support and supply of material for our studies. This work was supported by the Mass Spectrometry Based Proteomics Facility in Uppsala.

## Author Contributions

**Conceptualization:** Felix Sellberg, Fanny Fredriksson, Thomas Engstrand, Tim Melander Bowden, Folke Knutson, David Berglund.

**Data curation:** Felix Sellberg.

**Formal analysis:** Felix Sellberg, Fanny Fredriksson.

**Investigation:** Felix Sellberg, Fanny Fredriksson, Jaan Hong, Folke Knutson, David Berglund.

**Methodology:** Felix Sellberg, Jaan Hong, Folke Knutson.

**Project administration:** Felix Sellberg, Thomas Engstrand, Tim Melander Bowden, Bo Nilsson, Jaan Hong, Folke Knutson.

**Resources:** Thomas Engstrand, Tim Melander Bowden, Bo Nilsson, Jaan Hong, Folke Knutson, David Berglund.

**Supervision:** Fanny Fredriksson, Thomas Engstrand, Tim Melander Bowden, Bo Nilsson, Jaan Hong, Folke Knutson, David Berglund.

**Writing – original draft:** Felix Sellberg, Fanny Fredriksson, Jaan Hong, David Berglund.

**Writing – review & editing:** Felix Sellberg, Fanny Fredriksson, Thomas Engstrand, Tim Melander Bowden, Bo Nilsson, Jaan Hong, Folke Knutson.

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
