## [Decision Letter · Decision Letter 0]

25 Sep 2019

PONE-D-19-23341

Polyvinylalcohol-carbazate (PVAC) reduces red blood cell hemolysis

PLOS ONE

Dear Dr Sellberg,

Thank you for submitting your manuscript to PLOS ONE. After careful consideration, we feel that it has merit but does not fully meet PLOS ONE’s publication criteria as it currently stands. Therefore, we invite you to submit a revised version of the manuscript that addresses the points raised during the review process.

I normally wish to have more than one reviewer; however to provide a timely response, and the quality of review, I made my decision on one review.  Please address all of the issues raised. 

We would appreciate receiving your revised manuscript by Nov 09 2019 11:59PM. To enhance the reproducibility of your results, we recommend that if applicable you deposit your laboratory protocols in protocols.io, where a protocol can be assigned its own identifier (DOI) such that it can be cited independently in the future. For instructions see: http://journals.plos.org/plosone/s/submission-guidelines#loc-laboratory-protocols

We look forward to receiving your revised manuscript.

Kind regards,

Jeffrey Chalmers, Ph.D.

Academic Editor

PLOS ONE

Journal Requirements:

3. Thank you for your ethics statement:  For the whole blood chamber model whole blood was collected from healthy donors under permission of Regional Ethical Review Board (DNR 2008/264).

a)Please amend your current ethics statement to confirm that your named institutional review board or ethics committee specifically approved this study.

5. We note that you have a patent relating to material pertinent to this article. Please provide an amended statement of Competing Interests to declare this patent (with details including name and number), along with any other relevant declarations relating to employment, consultancy, patents, products in development or modified products etc. Please confirm that this does not alter your adherence to all PLOS ONE policies on sharing data and materials, as detailed online in our guide for authors http://journals.plos.org/plosone/s/competing-interests by including the following statement: "This does not alter our adherence to  PLOS ONE policies on sharing data and materials.” If there are restrictions on sharing of data and/or materials, please state these. Please note that we cannot proceed with consideration of your article until this information has been declared.

Additional Editor Comments (if provided):

Reviewers' comments:

Reviewer's Responses to Questions

**Comments to the Author**

1. Is the manuscript technically sound, and do the data support the conclusions?

Reviewer #1: Yes

2. Has the statistical analysis been performed appropriately and rigorously? 

Reviewer #1: Yes

3. Have the authors made all data underlying the findings in their manuscript fully available?

Reviewer #1: Yes

4. Is the manuscript presented in an intelligible fashion and written in standard English?

Reviewer #1: Yes

5. Review Comments to the Author

Reviewer #1: The manuscript, “Polyvinylalcohol-carbazate (PVAC) reduces red blood cell hemolysis” assesses whether PVAC is capable of inhibiting hemolysis in stored red blood cell units. RBC units with PVAC exhibited higher aldehyde binding and osmotic stability, resulting in reduced hemolysis. The authors postulate that PVAC administration may prevent the propagation of damage by oxidative byproducts. Further, PVAC appears to have a membrane stabilizing effect per hypotonic studies in the manuscript. Given that no toxicity has been observed with PVAC, the aldehyde scavenger represents a promising additive agent for RBC storage.

While the manuscript describes the methods in detail, the background could use more insight into current understanding of hemolysis during storage and relation to contemporary storage protocols. Further, the mechanism of carbonylation and any preventative pathways should be introduced. The methods section is thoroughly outlined, though the first section should be reworked to lay the foundation for later topics so that the section is easier to follow. The discussion ties together ideas and addresses any concerns with PVAC administration by citing toxicity data, though future studies will be necessary. Content feedback and grammatical edits may be found below.

1. Content Feedback

1.1. Page 3, Line 49: The terminology “such anti-hemolytic agents” should not be used here, unless these agents, or the notion thereof, are directly introduced in the preceding sentence.

1.2. Page 3, Line 58: It is stated that hemolysis shortens the possible length of storage. This statement may be bolstered with specification of contemporary RBC storage parameters and how these are contingent upon in-bag hemolysis.

1.3. Page 3, Line 59: How does carbonylation occur? Are there mechanisms in vivo that prevent carbonylation and/or its propagation?

1.4. Page 5, Line 86: Please expand upon “desired buffer and concentration.” While covered later in the methods, expanding upon this (i.e. listing the buffer options) would make this section more understandable and enhance cohesiveness of the methods section as a whole. Components should at least be noted for frame of reference.

1.5. Page 5, Line 87: The sentence states “the whole blood chamber model,” but this model has not yet been introduced. In the same vein of 1.4, adding detail here will help with understandability of the section as a whole.

1.6. Page 5, Line 88: By which metrics was donor health assessed?

1.7. Page 5, Line 95: Why was a second addition of PVAC employed for only the warm storage group, and not the cold storage group, following one week?

1.8. Page 6, Line 109: Please specify which signature corresponds to 540 nm (Q-band) and, thus, why absorbance at this wavelength was us used in analyses.

1.9. Page 8, Line 158: Please introduce what C3a is and why it is significant for assessment.

1.10. Page 11, Line 226: What is DNR 2008/264?

1.11. Page 13, Line 284: For figure four, the Y-axis reads “free aldehydes (%),” while the caption reads “free acetaldehyde.” Given that other aldehydes (i.e. MDA) may be present in the sample with storage age, perhaps the Y-axis should specify “acetaldehydes.”

1.12. Page 14, Line 332: What additional stresses are present at room temperature to cause additional hemolysis? Please specify. This point may further be driven by introducing contemporary storage parameters and why cold storage is instituted (i.e. how hemolysis mechanisms are altered at lower temperatures) in the background section.

2. Grammatical Edits

2.1. Page 3, Line 44: Prior to using the acronym “RBCs” in the body text, please introduce this in parentheses following “red blood cells.” In this section, “RBCs” is used, but throughout the body “RBC” is later used to refer to plural erythrocytes. These uses need be altered to “RBCs” for clarity.

2.2. Page 3, Line 46: Please change “haemolytic” to “hemolytic” for consistency with other uses of hemolysis throughout the manuscript.

2.3. Page 3, Line 47: A comma needs to be added prior to “where toxins lyse the RBC” for readability.

2.4. Page 3, Line 49: Please change “haemolytic” to “hemolytic” (see 2.2).

2.5. Page 4, Line 82: Both uses in of “is” here should be changed to “are.”

2.6. Page 6, Line 108: Minor typo of “minuntes.”

2.7. Page 8, Line 172: Should “immune precipitation” read “immunoprecipitation” here?

2.8. Page 12, Line 257: Please change the comma following “spectra” to a period.

2.9. Page 13, Line 280: Please change the comma following “acetaldehyde” to a period.

2.10. Page 13, Line 290: Please begin the sentence with “blood compatibility,” or some other terminology to complete this sentence.

2.11. Suppl. Line 6: “a appropriate” should be changed to “an appropriate.”

2.12. Suppl. Line 28: Please change “ice cold” to “ice-cold” for consistency with previous usage.

2.13. Suppl. Line 119: “none-modified” should be change to “unmodified.”

2.14. Suppl. Line 149: For consistency with other uses of “hem-“ in the main text, please change “haemoglobin” to “hemoglobin.”

2.15. Suppl. Line 154: See 2.14.

6. PLOS authors have the option to publish the peer review history of their article (what does this mean?). If published, this will include your full peer review and any attached files.

Reviewer #1: No

---

## [Author Response · Author response to Decision Letter 0]

3 Nov 2019

Response to reviewers

1. Content Feedback

1.1. Page 3, Line 49: The terminology “such anti-hemolytic agents” should not be used here, unless these agents, or the notion thereof, are directly introduced in the preceding sentence.

We agree with the comment and have rephrased the statement accordingly. 

1.2. Page 3, Line 58: It is stated that hemolysis shortens the possible length of storage. This statement may be bolstered with specification of contemporary RBC storage parameters and how these are contingent upon in-bag hemolysis.

We agree that this would strengthen the introduction to our work. We have expanded the introduction to include more information about the storage lesion including the role of carbonylation and how it relates to contemporary RBC storage.

1.3. Page 3, Line 59: How does carbonylation occur? Are there mechanisms in vivo that prevent carbonylation and/or its propagation?

Please see answer to 1.2 

1.4. Page 5, Line 86: Please expand upon “desired buffer and concentration.” While covered later in the methods, expanding upon this (i.e. listing the buffer options) would make this section more understandable and enhance cohesiveness of the methods section as a whole. Components should at least be noted for frame of reference.

We appreciate the comment by the reviewer and we have revised accordingly. 

1.5. Page 5, Line 87: The sentence states “the whole blood chamber model,” but this model has not yet been introduced. In the same vein of 1.4, adding detail here will help with understandability of the section as a whole.

We agree that this sentence was out of context and it has been rephrased to reference biocompatibility. 

1.6. Page 5, Line 88: By which metrics was donor health assessed?

“Healthy donors” are assessed using a screening process after which their blood donations can be used for clinical transfusion. This is a rigorous process that includes medical and laboratory assessments that determine the health status of the donor and that follow internationally established standards. A brief statement has been added in the section on RBC preparation. 

1.7. Page 5, Line 95: Why was a second addition of PVAC employed for only the warm storage group, and not the cold storage group, following one week?

The rationale for multiple additions in the warm storage group is that we expected a greater degree of hemolysis during warm storage, which was based on knowledge from laboratory and clinical practices utilizing cold storage to limit hemolysis. The lowering of temperature leads to a slower? metabolism which lowers oxidative stress and lysis of RBC. This was further confirmed in our study where the increase in temperature increased the hemolysis. The materials and methods has been edited to explain the rationale. 

1.8. Page 6, Line 109: Please specify which signature corresponds to 540 nm (Q-band) and, thus, why absorbance at this wavelength was used in analyses.

Depending on the oxidative state of hemoglobin the absorbance spectrum changes. RBC stored in the presence of oxygen are generally in the state of oxyhemoglobin which has an absorbance peak of approximately 540 nm. We confirmed this in our study by measuring the absorbance spectrum of hemoglobin between 300 and 700 nm (in sup Fig 5). We subsequently determined that 540 nm was an appropriate wavelength for detection also in our investigations. In materials and methods we have added a paragraph to clarify the wavelengths at which hemoglobin was measured. 

1.9. Page 8, Line 158: Please introduce what C3a is and why it is significant for assessment.

In materials and methods we have added a description of C3a, which is a molecule formed in the complement cascade that signals inflammation. In the whole blood model, increasing levels of C3a is an appropriate marker for general complement activation. 

1.10. Page 11, Line 226: What is DNR 2008/264?

This is an identification number used by the Swedish Regional Ethical Review Board. We have clarified this in our ethical statement in the materials and methods and added the statement to our supporting information. 

1.11. Page 13, Line 284: For figure four, the Y-axis reads “free aldehydes (%),” while the caption reads “free acetaldehyde.” Given that other aldehydes (i.e. MDA) may be present in the sample with storage age, perhaps the Y-axis should specify “acetaldehydes.”

We appreciate this astute observation and have clarified that we are indeed measuring not only acetaldehyde but in fact all free aldehydes. The method used relies on the enzyme acetaldehyde dehydrogenase and measures reduction of the substrates for the enzyme; we have also tested and confirmed that the method works with for example MDA. 

1.12. Page 14, Line 332: What additional stresses are present at room temperature to cause additional hemolysis? Please specify. This point may further be driven by introducing contemporary storage parameters and why cold storage is instituted (i.e. how hemolysis mechanisms are altered at lower temperatures) in the background section.

We have added additional information in the introduction/background regarding contemporary storage parameters. Please also see our response to point 1.2.

2. Grammatical Edits

2.1. Page 3, Line 44: Prior to using the acronym “RBCs” in the body text, please introduce this in parentheses following “red blood cells.” In this section, “RBCs” is used, but throughout the body “RBC” is later used to refer to plural erythrocytes. These uses need be altered to “RBCs” for clarity.

We have changed the abbreviation so that only RBCs is used throughout the manuscript, since we only refer to red blood cells in the plural form. 

2.2. Page 3, Line 46: Please change “haemolytic” to “hemolytic” for consistency with other uses of hemolysis throughout the manuscript. 

2.3. Page 3, Line 47: A comma needs to be added prior to “where toxins lyse the RBC” for readability.

2.4. Page 3, Line 49: Please change “haemolytic” to “hemolytic” (see 2.2).

2.5. Page 4, Line 82: Both uses in of “is” here should be changed to “are.”

2.6. Page 6, Line 108: Minor typo of “minuntes.”

2.7. Page 8, Line 172: Should “immune precipitation” read “immunoprecipitation” here?

2.8. Page 12, Line 257: Please change the comma following “spectra” to a period.

2.9. Page 13, Line 280: Please change the comma following “acetaldehyde” to a period.

2.10. Page 13, Line 290: Please begin the sentence with “blood compatibility,” or some other terminology to complete this sentence.

2.11. Suppl. Line 6: “a appropriate” should be changed to “an appropriate.”

2.12. Suppl. Line 28: Please change “ice cold” to “ice-cold” for consistency with previous usage.

2.13. Suppl. Line 119: “none-modified” should be change to “unmodified.”

2.14. Suppl. Line 149: For consistency with other uses of “hem-“ in the main text, please change “haemoglobin” to “hemoglobin.”

2.15. Suppl. Line 154: See 2.14.

All grammatical edits (2.2 – 2.15) suggested by the reviewer have been corrected. Furthermore, an experienced English proof reader has reviewed the manuscript and made additional edits for readability.

---

## [Editor Report · Decision Letter 1]

13 Nov 2019

Polyvinylalcohol-carbazate (PVAC) reduces red blood cell hemolysis

PONE-D-19-23341R1

Dear Dr. Sellberg,

We are pleased to inform you that your manuscript has been judged scientifically suitable for publication and will be formally accepted for publication once it complies with all outstanding technical requirements.

With kind regards,

Jeffrey Chalmers, Ph.D.

Academic Editor

PLOS ONE
---

## [Editor Report · Acceptance letter]

22 Nov 2019

PONE-D-19-23341R1 

Polyvinylalcohol-carbazate (PVAC) reduces red blood cell hemolysis 

Dear Dr. Sellberg:

I am pleased to inform you that your manuscript has been deemed suitable for publication in PLOS ONE. Congratulations! Your manuscript is now with our production department. 

With kind regards,

on behalf of

Dr. Jeffrey Chalmers 

Academic Editor

PLOS ONE